# Solid Lipid Nanoparticles: Applications and Prospects in Cancer Treatment

**DOI:** 10.3390/ijms24076199

**Published:** 2023-03-24

**Authors:** Durgaramani Sivadasan, Kalaivanan Ramakrishnan, Janani Mahendran, Hariprasad Ranganathan, Arjunan Karuppaiah, Habibur Rahman

**Affiliations:** 1Department of Pharmaceutics, College of Pharmacy, Jazan University, Jazan 45142, Saudi Arabia; 2Department of Pharmaceutics, PSG College of Pharmacy, Coimbatore 641004, TN, India; 3Department of Pharmaceutics, College of Pharmacy, Sri Ramakrishna Institute of Paramedical Sciences, Coimbatore 641002, TN, India; 4Department of Pharmaceutical Analysis, PSG College of Pharmacy, Coimbatore 641004, TN, India; 5Department of Pharmaceutics, Karpagam College of Pharmacy, Coimbatore 641032, TN, India

**Keywords:** lipid nanoparticle, cancer therapeutics, drug delivery mechanism

## Abstract

Recent advancements in drug delivery technologies paved a way for improving cancer therapeutics. Nanotechnology emerged as a potential tool in the field of drug delivery, overcoming the challenges of conventional drug delivery systems. In the field of nanotechnology, solid lipid nanoparticles (SLNs) play a vital role with a wide range of diverse applications, namely drug delivery, clinical medicine, and cancer therapeutics. SLNs establish a significant role owing to their ability to encapsulate hydrophilic and hydrophobic compounds, biocompatibility, ease of surface modification, scale-up feasibility, and possibilities of both active and passive targeting to various organs. In cancer therapy, SLNs have emerged as imminent nanocarriers for overcoming physiological barriers and multidrug resistance pathways. However, there is a need for special attention to be paid to further improving the conceptual understanding of the biological responses of SLNs in cancer therapeutics. Hence, further research exploration needs to be focused on the determination of the structure and strength of SLNs at the cellular level, both in vitro and in vivo, to develop potential therapeutics with reduced side effects. The present review addresses the various modalities of SLN development, SLN mechanisms in cancer therapeutics, and the scale-up potential and regulatory considerations of SLN technology. The review extensively focuses on the applications of SLNs in cancer treatment.

## 1. Introduction

Cancer is recognized as one of the world’s most deadly diseases in its different forms. The capacity of treatments to reach specific intracellular and intercellular targets while reducing their accumulation at nonspecific areas is critical to treat cancer. Chemotherapy performed via conventional drug administration is the most comprehensive cancer treatment, but it has a number of drawbacks such as lower drug solubility and specificity, lower therapeutic index, and increased toxicity. The resistance of malignant cells to chemotherapy drugs is another hurdle in the way of treatment, and it is recognized as multidrug resistance (MDR), causing resistance to a wide range of medications [1]. Solid lipid nanoparticles (SLNs) have evolved as a potential nanodelivery system (nanocarriers) in cancer treatment. The colloidal carriers such as emulsions, liposomes, and polymeric micro and nanoparticles were apparently replaced by the development of SLNs (1991) [2]. SLNs are submicron colloidal systems that contain physiological lipids dispersed in an aqueous surfactant solution or water but remain solid in the body environment. When compared to traditional colloidal carriers, SLNs have lower toxicity, a larger surface area, prolonged drug release, higher cellular absorption, and the ability to improve drug solubility and bioavailability [3]. The matrix type and the drug’s position determine the drug release of the formulation. SLNs composed of biodegradable and biocompatible (e.g., physiological lipids or lipid molecules) materials may incorporate both hydrophilic and lipophilic bioactives and act as a potential choice for a targeted drug delivery system [4]. The drug is dispersed or dissolved in the hydrophobic solid core, which has a monolayer phospholipid coating. The particle size after drug encapsulation ranges from nanoscale to submicron scale (50–1000 nm); the synthesis of nanoparticles does not require the use of organic solvents, and the process (e.g., high-pressure homogenization) may be accomplished at reduced cost and easily scaled up [5]. The structure of an SLN is depicted in Figure 1. Various compositions of SLNs are explained in Figure 2.

### 1.1. Pros

Due to the nanoscale size range of SLNs, reticuloendothelial system (RES) cells were not able to absorb SLNs, allowing them to bypass filtration by the spleen and liver.The incorporated drugs will have better stability.Hydrophilic and lipophilic drugs can both be incorporated.Increased bioavailability of substances that are poorly water-soluble.Easy sterilization and scale-up process.The immobilization of drug molecules within solid lipids protects sensitive pharmaceuticals from various factors such as photochemical, oxidative, and chemical degradation, as well as reducing the risk of drug leakage.Drying using lyophilization is feasible.It is possible to obtain a targeted and controlled drug release.Biocompatible and biodegradable compositional ingredients.Lower toxicity, a larger surface area, prolonged drug release, higher cellular absorption, and the ability to improve drug solubility and bioavailability.

### 1.2. Cons

Low loading efficiency.In storage conditions, polymorphic transition leads to drug leakage.The dispersions have relatively high water content (70–99.9%).

### 1.3. Categories of SLNs

SLNs are categorized into three classes depending on the composition of the active component and lipid, the nature and concentration of surfactants, the drug’s solubility in melted lipid, the type of manufacturing, and the production temperature [4,6]. The various types of SLNs are explained in Table 1 and Figure 3.

SLN Type I or homogeneous matrix model.SLN Type II or drug-enhanced shell model.SLN Type III or drug-enhanced core model.

## 2. Formulation Techniques

The various formulation modalities of the SLN formulations are briefly elaborated in Figure 4.

### 2.1. Delivery Mechanisms of SLNs

The lipid and its composition impact the drug’s release mechanism from the SLN matrix, a versatile or dual release system (i.e., immediate release and sustained release) as the drug is either incorporated into the matrix or on the surface and drug attached to the surface will dissociate, causing a release effect; thereafter, depending on the lipid content, the matrix will erode or degrade, and the drug is released in a controlled manner [7]. Burst drug release from SLNs can be triggered by temperature or a large quantity of surfactant. The SLN drug entrapment model affects drug distribution; for example, with the drug-enriched shell model, drug release is faster, because high enthalpy dissolves the drug and the temperature affects the drug’s solubility in water, which leads to drug deposition on the lipid matrix’s outer surface [8]. The melting of the lipid core induced rapid release of loaded 5-fluorouracil (>90%) at 39 degrees Celsius, whereas the solid core caused 22–34% drug release at 37 °C [9]. The drug release pattern of cholesterol-PEG-coated SLNs was studied; at pH 4.7, these particles release doxorubicin-loaded nanoparticles more rapidly than at pH 7.4. The accelerated release at low pH has been attributed to the loss of electrostatic interactions between the positively charged doxorubicin and the negatively charged lipid core lauric acid (protonation) [10].

#### 2.1.1. Passive Delivery

The enhanced permeability and retention effect allows tumor targeting due to the tumor microenvironment characteristics. In normal conditions, nanoparticle extravasation does not occur; however, the discontinuity of the vascular epithelium in the tumor region and the improper functioning of the lymphatic drainage system facilitate enhanced extravasation. Angiogenesis stimulates the formation of irregular blood vessels with discontinuous epithelium in tumor sites. The increased permeability is due to the discontinuities between epithelial cells, nanoparticles in the size range from 100 to 800 nm can flow across the interstitial space. Tumor tissues have a dysfunctional lymphatic system and insufficient lymphatic outflow, resulting in nanoparticle accumulation in the tumor tissue. The EPR effect influences molecular distribution through three related mechanisms: nanoparticle extravasation from blood arteries, nanoparticle diffusion into cancer tissue, and nanoparticle interaction with intracellular or extracellular targets in the tumor microenvironment [11,12]. The passive delivery mechanism is shown in Figure 5.

#### 2.1.2. Active Delivery

Target molecules such as receptors or transporters that are overexpressed on the surface of tumor cells are recognized by active delivery. Pharmacological agents can be administered selectively to tumor cells, reducing disruption and limiting unwanted side effects by modifying the surface of nanoparticles. Surface modification of nanoparticles allows ligands of interest and results in a high binding high affinity to the transporter. Surface modification has minimal impact on the biodistribution characteristics of nanoparticles [13]. In HeLa cervix and MCF-7 breast cancer lines, SLNs loaded with paclitaxel and complexed with hyaluronic acid and pluronic acid (an inhibitor of the efflux transporter P-glycoprotein) have the ability to overcome drug tolerance and impair cell viability. Furthermore, as compared to SLNs without hyaluronic acid and free drug, these SLNs significantly improve drug concentration and efficacy in tumor tissues in mice by active delivery [14]. Tetraiodothyroacetic acid modification would allow drugs to be transported to the surface of tumors, improving the hyaluronic acid’s active delivery capacity. DocetaxelSLNs loaded with multiple ligandsallow targeted delivery and are delivered to the surface of tumors after modification with tetraiodothyroacetic acid, increasing active delivery capacity, andin vitro experiments revealed increased incorporation and decreased viability in B16F10 mouse melanoma cells (expressing v3 and CD44) andtumor growth inhibition [15].

### 2.2. Codelivery Mechanisms

To overcome drug resistance in tumor cells, a combination of two distinct nanoparticle compounds is used: an anticancer drug and an agent that acts against MDR mechanisms. Inhibitory substances, such as small interfering RNA (siRNA)for inhibiting ABC transporter gene expression and microRNAs (miRNAs), allow post-transcriptional gene regulation [16]. Paclitaxel, an anticancer drug, and Hsp90 (inhibitor) combined with SLN drug delivery (synergic effect) enhanced anticancer activity, lowering the size and weight of gastric tumors in mice, as well as cell viability in several human gastric cell lines [17]. SLNs were loaded with polyethyleneglycol-distearoyl-phosphatidylethanolamine (PEG-DSPE)loaded and functionalized via TAT (trans-activating transcriptional activator), a peptide that enables the nanocarriers to penetrate cells more effectively. Paclitaxel and cisplatin were used for administration and were conjugated with tocopherol succinate, a vitamin E derivative. Functionalization and codelivery revealed a synergistic effect by increasing cellular absorption in HeLa cells (in vitro) and anticancer efficacy (in vivo), as well as reducing the number of cervical tumors in mice [18].

### 2.3. Antiadhesive Mechanism

Cholesteryl butyrate promotes cancer cell adhesion to the endothelium and is important in metastatic spread dissemination. SLNsare used to improve cancer cell adhesion. A computerized micro-imaging system was used to measure adhesion after cholesteryl butyrate SLNs were incubated with cancer or endothelial cells. The Boyden chamber invasion test and the scratch “wound healing” assay were used to identify migration. The expression of ERK (extracellular regulatory kinase) and p38 MAPK (mitogen-activated protein kinase) was examined using a Western blot. It was discovered that SLNs can operate as an antimetastatic medication, which adds a new mechanism to the drug’s antitumor effect [19].

## 3. Possible Routes of Administration for SLNs

### 3.1. Topical Route

Because of inherent biocompatibility, SLNs are frequently employed in topical treatments. With increased exclusivity and hydration of the stratum corneum, lipophilic drugs loaded with SLNs had better skin penetration than free drugs. After application, SLNs progressively convert into the stable polymorph, allowing for long-term release. Drugs are released in a controlled mannerfrom the SLNs if the polymorphic transitions are regulated by the addition of surface-active compounds [20]. Vitamin E, tocopherol acetate, retinol, ascorbyl palmitate, clotrimazole, triptolide, phodphyllotoxin, and the nonsteroidal antiandrogen RU 58841 have all been examined with SLNs for topical administration [21].

### 3.2. Pulmonary Route

The pulmonary route provides the ability to transfer medications to the systemic circulation in a non-invasive way via a device or inhaler, avoiding first-pass metabolism, treating lung diseases, and improving medication absorption and transport efficacy in alveolar macrophages [20]. SLN nebulization is a relatively new field of study. The absorption of particles in the respiratory system is greatly facilitated by lymphatic drainage. SLNsare used as carriers for anticancer medications or peptide therapeutics to enhance their bioavailability in lung cancer treatment. The biodistribution of radio-labeled SLNsthat were inhaled was studied, and reports revealed a considerable uptake of the radio-labeled SLN into the lymphatics after inhalation [22]. Antitubercular drugs (rifampicin, isoniazid, and pyrazinamide) were embedded into a variety of SLN formulations ranging from 1.1 to 2.1 μm in size, and the formulations were delivered through the pulmonary route to guinea pigs by mouth via a nebulizer. This delivery method showed improved drug bioavailability and reduced dosage frequency, resulting in improved pulmonary tuberculosis therapy [23].

### 3.3. Oral Route

Orally, SLNs can be administered as a suspension or as a solid dose form such as a tablet, capsule, or dry powder. SLNs loaded with lopinavir were produced using a hot self-nano-emulsification method to improve the drug’s bioavailability, and due to the drugSLNs’ high intestinal lymphatic absorption, lopinavir oral bioavailability was significantly increased [24]. Nanoparticles are absorbed through the mucosa of the gut by a variety of mechanisms, notably Peyer’s patches, intracellular absorption, and the paracellular route. The release of SLNs from spherical pellets for oral delivery was studied [25]. Antitubercular drugs (rifampicin, isoniazid, and pyrazinamide) were entrapped in polyvinyl alcohol-coated SLNs after a single oral injection to mice, and therapeutic drug concentrations were maintained in plasma for 8 days and in organs (lungs, liver, and spleen) for 10 days [23]. Surface modification with PEG-stearate significantly improved the stability and resistance to lipolytic enzymes. A study coating particles with hydrophilic chemicals such as PEG for oral calcitonin delivery via SLNs shows that the coating’s composition might impact surface association, and, as a result, the peptide is released immediately. This method has the potential for continuous administration of the linked peptide and increased drug bioavailability [26]. The stability, drug release, and bioavailability of risperidone-loaded SLNs for oral administration were studied [27]. SLNs were loaded with rifampicin to inhibit drug hydrolysis in acidic pH. These methods prevent drug degradation and therapeutic failure [28].

### 3.4. Parenteral Route

SLNs, made up of physiologically well-tolerated components that possess large storage capacities following lyophilization and/or sterilization, are ideal for systemic distribution. When administered intravenously, SLNs are tiny enough to pass through the microvascular system and impede macrophage absorption in the presence of the hydrophilic coating. The most researched route of administration for SLNs is intravenous (i.v.) injection, especially for targeted delivery. The pharmacokinetics and biodistribution of camptothecin-loaded SLNs delivered intravenously in mice were studied. SLNs showed improved AUC/dose and mean residence times (MRTs) in the brain compared to the free drug. The largest concentration of SLNs in the brain compared to the free drug among the studied organs showed that this carrier has brain-targeting potential [29]. A cationic SLN binds genes directly via electrostatic interactions and is effective in cancer treatment via targeted gene therapy. The composition can alter the charge of particles, allowing the binding of oppositely charged molecules [30].

### 3.5. Ocular Route

For ocular delivery, SLNs presented good penetration properties when compared to typical ophthalmic solutions; the drug’s distribution into the ocular mucosa can be prolonged or regulated, increasing the drug’s retention time in the pre-corneal area [31]. The parameters for SLNs’ ocular distribution should include ocular compatibility (Draize rabbit eye test), sterility, isotonicity, and pH value (similar to lachrymal fluid) [32]. The biocompatibility and mucoadhesive properties of SLNs in ocular drug targeting prolong the drug’s interaction with the ocular mucosa and the drug’s corneal residence time [33]. In rabbit eyes, tobramycin was loaded in SLNsthat were used as a carrier for ocular administration. The concentration of the drug in the aqueous humor was tested for up to six hours, and it showed improved drug bioavailability in the aqueous humor [34].

## 4. Characterization Parameters of Solid Lipid Nanoparticles

There are numerous characterization parameters influencing the efficacy of SLNs; the parameters are presented in Figure 6.

### Current Patents (Source: WIPO, Google Patents, Lens)

Many studies are ongoing in the development of SLNs for various treatment approaches. The current patents on solid lipid nanoparticles are presented in Table 2. The various clinical trials ongoing for the specific use of SLNs in cancer treatment are described in Table 3. There are few products available in the market based on solid lipid nanoparticle technology. The details are given in Table 4.

## 5. Applications of Solid Lipid Nanoparticles in Cancer

### 5.1. Breast Cancer

Breast carcinoma is the most common malignancy in women, and its rate of incidence is rising gradually over time. Insufficient drug concentrations approaching the carcinoma, prompt excretion, systemic toxicity, and side effects are all significant impediments to effective breast cancer chemotherapies. SLNs in the treatment of breast cancer have the potential to overcome prevailing chemotherapeutic restrictions, as well as the issues related to conventional chemotherapy and MDR [35]. Chemoresistance or MDR can be induced by either one or two mechanisms: physical impairment of drug delivery to the tumor site (e.g., poor absorption, enhanced metabolism/excretion, and/or impaired drug diffusion into the tumor mass)and intracellular mechanisms which elevate apoptosis. SLNs are effective in targeting tumor vehicles because of the EPR effect’s passive targeting characteristics. Paclitaxel (PAX)-loaded SLNs were investigated, and it was shown that the modified SLNs were stable and reproducible [36]. The dimethyl sulfoxide solubilization Cremophor EL vehicles were studied for effectiveness against MCF-7 drug-resistant and drug-sensitive cells, and SLNs loaded with paclitaxel were targeted against drug-resistant breast cancer cells. High IC50 concentration was reported in drug-resistant cells, according to a study on concentration-dependent cytotoxicity and enhanced cellular uptake, particularly in drug-resistant cells, indicating their potency in evading multidrug resistance pathways in breast cancer cells [3]. SLNs incorporated with curcumin present increased cellular drug absorption capacity, and the use of curcumin carriers to combat MDA-MB-231 (breast cancer cell line) shows their potency against cancer and a decrease in cell viability and an elevation in apoptotic cells when compared to dimethyl sulfoxide-diluted curcumin [37]. Methotrexate-loaded SLNs were ferrous-functionalized to achieve active tumor targeting, resulting in an enhanced cytotoxic effect in MCF-7 breast cancer cells and in rats with induced breast cancer; they showed enhanced drug concentration in tumor sites and improved anticancer activity [38].

### 5.2. Lung Cancer

Lung cancer (LuC) is the most frequently diagnosed cancer in both men and women and is the main cause of cancer-related death worldwide. Despite the fact that chemotherapy and radiation are effective in treating lung cancer, a vast proportion of patients develop a relapse of the illness that is more resistant to subsequent treatment. As a result, in order to enhance the prognosis of this kind of cancer, a novel therapeutic strategy is required. A study of SLNs loaded with anticancer compound naringenin reported an increased cellular uptake pattern following intratracheal administration of naringenin-loaded SLNs in rats according to the drug’s pharmacokinetic properties, such as mean residence time and maximum plasma concentration.SLNs can be administrated through the lungs by inhalation, and paclitaxel loaded into SLNs coated with a polymer composed of folate-poly (ethylene glycol) and chitosan lowered the IC50 value in vitro against M109HiFR lung cancer cells and increased drug concentration in the lungs of healthy and sick mice [39]. A study of SLNs incorporated with erlotinib indicated that the free drug exhibited a lower cytotoxic effect than the drug encapsulated in SLNs, and the system exhibited adequate aerosol dispersion capability, indicating that it could be exploited for pulmonary delivery [40].

### 5.3. Colon Tumor

SLNs have the potential to be an effective treatment for colon cancers. SLNs were found to inhibit cell growth more than free fatty acid in HT-29 and GCT116 adenocarcinoma cells by increased apoptotic activation [41]. Oxaliplatin incorporated in SLNs with folic acid resulted in increased chemotherapeutic activity against the HT-29cell line compared to the free drug and non-functionalized SLNs [42]. For the treatment of CRC, SLNs containing 5-fluorouracil (5-FU) have greater anticancer activity than pure 5-FU [43].

### 5.4. Prostate Cancer

There has been an increase in the incidence of prostate cancer due to the inability to target therapies to neoplastic cells, and SLNs are reported to be effective at inhibiting prostate cancer cells (e.g., LNCap) as a drug delivery system. Studies indicate that adjusting the process parameters (e.g., pressure/temperature) and using different lipids increase the anticancer activityof SLNs loaded withretinoic acid (RTA); RTA-SLNs incubated in LNCap cell lines exhibited decreased cell viability and higher drug concentrations (e.g., 9.53% at 200 g/mL), but blank SLNs exhibited no cytotoxicity [44].

### 5.5. Liver Cancer

Liver cancer causes a significant amount of mortality, and treatments for liver cancer (LivC) are frequently restricted by poor drug physicochemical characteristics. Chemotherapy and targeted medicines such assorafenib have just a minimal effect on patient survival. Furthermore, radiation is unsuccessful in most cases, requiring the development of alternative therapeutic approaches. SPIONs were added to HepG2 cell line human hepatocyte carcinoma, and sorafenib-loaded SLNs exhibited a significant cytotoxic impact, although they were not as effective as the free drug. However, cellular absorption of SLNs and magnetic targeting tests demonstrated improved hepatocellular carcinoma therapy [45]. The use of linalool SLNs loaded with different formulations against HepG2 cell line human hepatocyte carcinoma and A549 lung adenocarcinoma reporter gene cell lines was studied;they showed strong antiproliferative activity inthe HepG2 human hepatocellular carcinoma cell line [46].

### 5.6. Brain Cancer

The use of SLNs is an effective nanoscale lipid-based approach for brain cancer drug delivery. The specific process of the delivery system passing through the BBB is unclear; it is thought that internalization is facilitated by endothelial cells (pinocytosis). The procedure of endocytosis/pinocytosis leads to the effective absorption of circulating plasma proteins onto the SLN surface [47]. SLNs incorporated with the drug indirubin showed an increased cytotoxic effect in acidic conditions in U-87 MG human cell line [48]. SLNs coated with apolipoprotein E (ApoE)showed active cellular uptake and increased accumulation in the brain (ApoE-SLN) [49,50]. Resveratrol functionalized with apolipoprotein E can be delivered into the brain by loading it into SLNs [51]. Andrographolide (AG) was delivered into the brain by loading it into SLNs and using Compritol 888 ATO as a solid lipid in healthy rats; the ability to cross the BBB was tested in vitro and in vivo [52].

## 6. Stability and Storage

The stability of SLNs is determined primarily by the lipid material and the concentration of surfactants, and the temperature optimization during preparation impacts thestability and storage. During preparation and storage, triglycerides undergo α (alpha), β’(beta prime), and β (beta) crystal alteration. SLNs were exposed to various destabilizing agents; they gelled, and their zeta potential was significantly reduced. The drug may hydrolyze in aqueous dispersion, and SLNs have a variety of stability issues. Drying is essential for the long-term storage of SLNs. Drying techniques include freeze drying, spray drying, and lyophilization. The electrospray technique was recently utilized to make SLNs, yielding a dry SLN powder immediately. The powder form of SLNs is greatly advantageous for drug administration as it can be incorporated into pellets, capsules, or tablets. Due to the nature of SLN formulations, their application is restricted; uncontrolled particle development via coagulation or agglomeration leads to extremely rapid “burst release”. Crystal lipid–lipid matrices in SLNs transport the loaded drug in its molecular form between fatty acid chains. The release of the loaded drug solution is induced by the uncontrolled, undesirable increase in the crystal structure during manufacture and storage [21,53].

## 7. Future Directions

Numerous nanomaterials have been studied extensively in cancer therapy [54]. SLNs are a novel drug delivery technology that provides a solution for a wide range of drug administration issues. As a carrier of colloidal drugs, SLNs incorporate the benefits of polymeric nanoparticles (NPs) and fat-based emulsions and have a few advantages, including the easy application of soluble and lipid-soluble drugs, adequate physical stability, low cost, and easy production. Nevertheless, it is notable that these nanoparticles overcome the problems encountered by traditional drug delivery systems and provide numerous advantages. SLNs can transport drugs with physicochemical incompatibility, drugs with a low pharmacokinetic profile, and thermolabile compounds to the target area. Delivery of proteins and peptides with high levels of efficacy and low toxicity can also be achieved through SLNs, which can act as a potential tool to combat cancer-related diseases. In the future, the development of surface-modified SLNs may be of tremendous importance for active and targeted delivery in various types of malignancies with several related resistance mechanisms, and an exceptionally vast number of nanotechnology-based medicines and diagnostics are making their way into clinical trials.

## 8. Conclusions

The pharmaceutical sector has entered a new era with SLNs. A combination of drugs and SLNs might be used to develop targeted cellular and tissue-specific clinical applications to reach maximum therapeutic effects with reduced side effects. By modulating passive, active, and co-transport mechanisms, SLNs improve cellular uptake of encapsulated drugs and are able to bypass biological barriers. The various techniques used to prepare SLNs were found to be as simple as those used to prepare other conventional drug carriers. SLNs can deliver drugs that are physicochemically incompatible, have poor pharmacokinetic profiles, or are thermolabile to the target site. SLNs can also be used to transport proteins and peptides with greater efficiency and lesser toxicity. Many SLN-related patents have already been filed; additional SLN-based delivery systems may soon be patented. During the development of pharmaceutical products, the stability of the drug nanoparticle remains a major challenge. Further research is needed to determine the structure and strength of SLNs at the cellular level, both in vitro and in vivo. The conceptual understanding of biological responses (adsorption/desorption processes, enzymatic degradation, agglomeration, and interaction with endogenous lipid carrier systems) to nanostructure-based drug delivery systems is crucial for cancer therapy in the future.

## Figures and Tables

**Figure 1 ijms-24-06199-f001:**
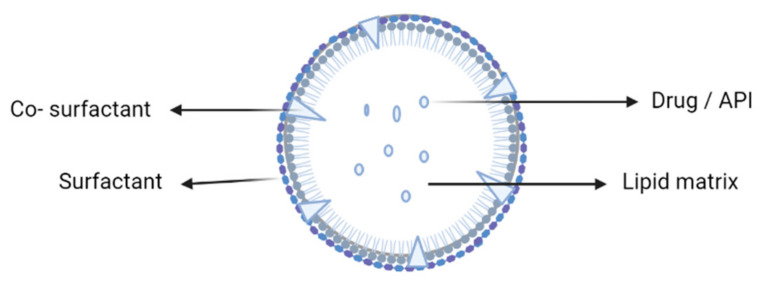
Structure of a solid lipid nanoparticle.

**Figure 2 ijms-24-06199-f002:**
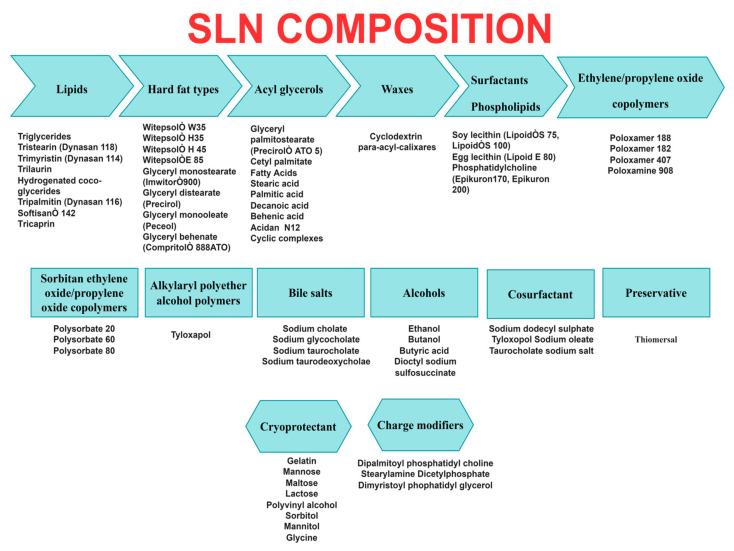
Ingredients used in SLN formulations.

**Figure 3 ijms-24-06199-f003:**
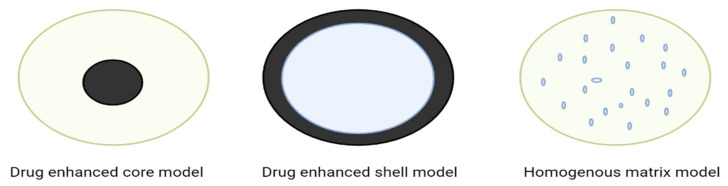
Types of solid lipid nanoparticles.

**Figure 4 ijms-24-06199-f004:**
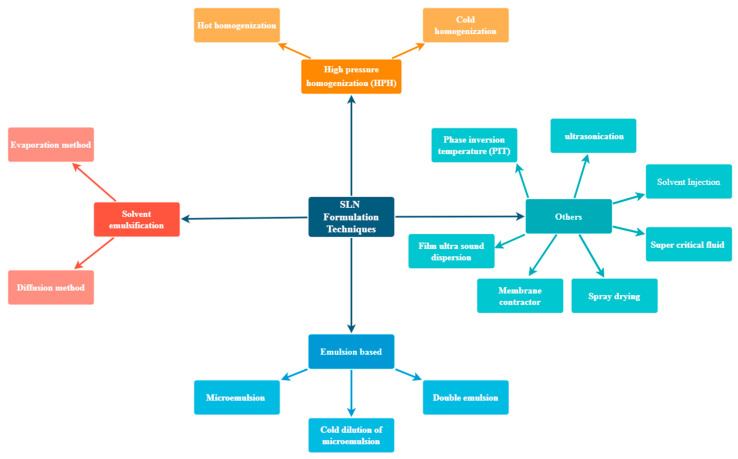
Various Formulation Techniques.

**Figure 5 ijms-24-06199-f005:**
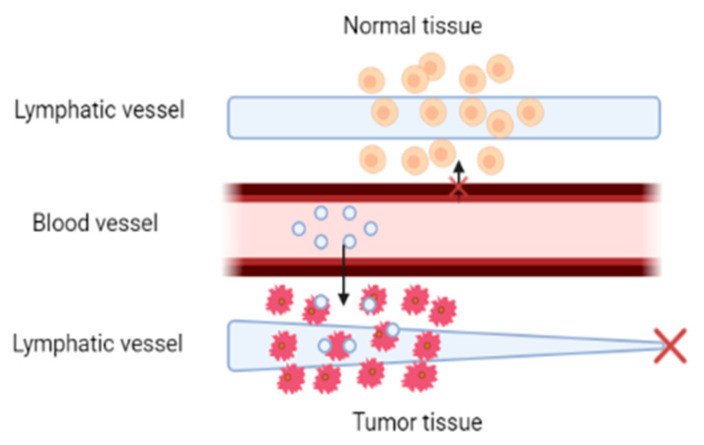
Passive delivery mechanism (adapted from [7]).

**Figure 6 ijms-24-06199-f006:**
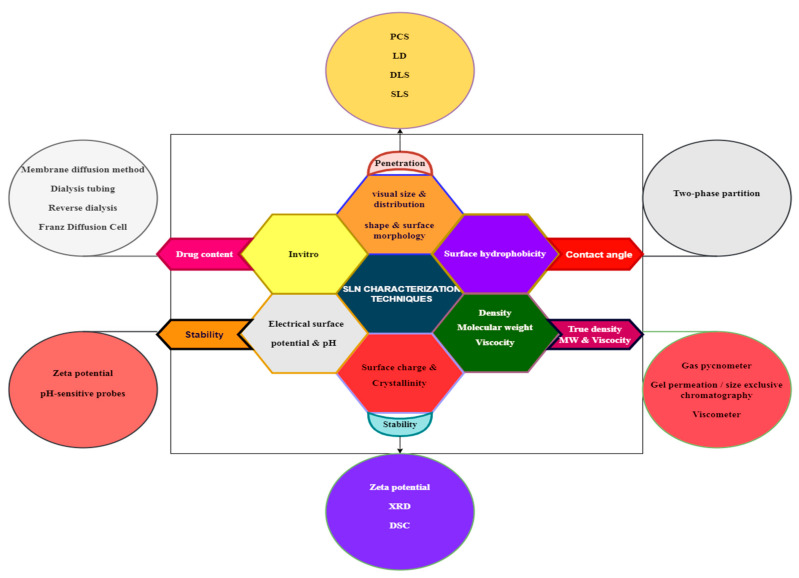
Characterization parameters.

**Table 1 ijms-24-06199-t001:** Types of SLN formulations.

SLN Type 1	SLN Type 2	SLN Type 3
Homogeneous matrix model	Drug-enhanced shell model	Drug-enhanced core model
Active pharmaceutical ingredient (API) is either molecularly distributed in the lipid core or exists as amorphous clusters.	The melted lipid has a low concentration of API.	In this method, a drug is solubilized in a lipid melt until it obtains saturation solubility.
This model is developed by using appropriate API and lipid ratios through high-pressure homogenization (HPH) at temperatures above the lipid’s melting point, or by employing the cold HPH approach.	Hothomogenization technique is used for the formulation. At the recrystallization temperature of the lipid, the lipid core is prepared.The drug partitions into the lipid phase when the dispersion is cooled.	Active ingredient is super-saturated in cold dispersion, allowing the drug to dissolve in the lipid, and the melted lipid is the drug’s precipitation.
No solubilizing agent is used.	This approach can be utilized to obtain a burst release of the API rather than a prolonged API release.	Further cooling induces the lipid to recrystallize, forming a membrane surrounding the existingcrystallized drug-enriched core.
Controlledrelease properties.	This approach can be utilized to obtain a burst release of the API rather than a prolonged API release.	Prolonged release properties.
Drug and lipid show strong interaction.	The drug is concentrated in the shell’s outer layer.	The drug is concentrated in the core.

**Table 2 ijms-24-06199-t002:** Current patents on SLNs.

Application Number	Title	Applicant/Inventor	Filling Date	Publication Date
US20210069121A1	Solid lipid nanoparticle for intracellular release of active substances and method for production the same	Christo Tzachev Tzachev	12 December 2017	11 March 2021
US10166187B2	Curcumin solid lipid particles and methods for their preparation and use	Christopher, Diorio John Lokhnauth	18 October 2017	8 February 2018
US 2020/0197360	Dispersion of Formononetin Solid Lipid Nanoparticles and Process for Its Preparation	Lakshmi Karunanidhi Santhana et al.	23 December 2018	13 April 2021
CN112870178A	Phenanthroindolizidine alkaloid derivative solid lipid nanoparticle composition	Lucy Liu et al.	29 November 2019	1 June 2021
US10780184B2	Fluorescent solid lipid nanoparticles composition and preparation thereof	BraccoImaging S.P.A.	31 January 2019	6 June 2019
WO 2021/133335	Therapeutic Use of a Ceramidase Inhibitor B13 and Its Nano Form in Lung Cancer Cells	Mehtap Kutlu Haticeet al.	18 December 2020	1 July 2021

**Table 3 ijms-24-06199-t003:** List of SLNs under clinical trials.

Trial/Identifier Number	Disease Type	Status	Phase	Study Completion Date
NCT02110563	Solid Tumors, Multiple Myeloma, etc.	Terminated	Phase 1	3 November 2016
NCT02110563	Solid Tumors, MultipleMyeloma, PNET, NHL, etc.	Terminated	Phase 1	3 November 2016
NCT02314052	Hepatocellular Carcinoma	Terminated	Phase1b /phase2	11 October 2016
NCT03323398	Advanced Malignancies	Recruiting	Phase 1	July 2019 (Anticipated)
NCT03323398 (History ofChanges for Study)	RefractorySolid Tumor Malignancies or Lymphoma	Recruiting	Phase1/phase2	October 2020
NCT03739931	TNBc, Refractory Solid Tumor Malignancies, etc.	Recruiting	Phase 1	31 July 2021
NCT03739931 (History ofChanges forStudy)	Relapsed/Refractory SolidTumor Malignancies or Lymphoma	Recruiting	Phase 1	November 2021
NCT03739931	TNBc, Refractory Solid Tumor Malignancies, etc.	Recruiting	Phase 1	July2021 (Anticipated)
NCT04675996	Solid Tumor	Recruiting	Phase 1	December 2024

**Table 4 ijms-24-06199-t004:** SLN-based products in the market.

Manufacturer	Brand Name	Drug/Active	Route of Administration	Field of Application
Bayer HealthCare Pharmaceuticals Inc.	Cipro	Ciprofloxacin	Oral	Pharmaceuticals
Boehringer	Mucosolvan Retard	Ambroxol	Oral	Pharmaceuticals
Yamanouchi	Nanobase	-	Dermal	Cosmeceuticals

## Data Availability

No new data were created or analyzed in this study. Data sharing is not applicable to this article.

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
