# Peer review of "Solid Lipid Nanoparticles: Applications and Prospects in Cancer Treatment"

_ijms, 2023, doi:10.3390/ijms24076199_

Round 1

Reviewer 1 Report

Dear Authors 

The manuscript is well written with recent scientific information except few minor corrections .

The comments are attached 

Reviewer 2 Report

1. Spell check must be done throughout the manuscript

2. Kindly provide an additional reference for the stability of SLNs

3. Check reference format uniformity 

4. Avoid fluorescent colors in the flow chart 

5. Any marketed products for cancer in SLN

6. Please check if your references are from journals indexed by standard agencies, such as SCI indexed (including SCI-expanded and Emerging SCI lists), Scopus, PubMed, etc.

Reviewer 3 Report

This manuscript investigated the role of Solid Lipid Nanoparticles in in cancer treatment.  I suggest a minor correction and require a detailed clarification. Correction to be addressed by the authors as follows: The abstract is not well organized, where the sentences are incomplete and no continuity is there. It would be feasible, if include the significance of the current study in the abstract. A brief description of how the authors selected information from the literature in the databases, as well as doses.
Authors should justify and expand the information on the biomedical application of Solid Lipid Nanoparticles, highlighting the main contribution in in vitro fields. Authors should specify the main experimental conditions used on the evidences from the literature. Where they briefly describe the most important data reported in the literature in a homogeneous manner and sequence reinforcing the relevance of Solid Lipid Nanoparticles as medicinal alternative.
The most significant  mechanism of action of this nanoparticles should be described and noticed more emphatically. Authors should discuss whether the use of these nanoparticles represents a solid alternative to existing commercial drugs or a source of new drugs.
Please add below studies to your manuscript in discussion section and also please discuss about possible toxicity of proposed nanomaterials.

DOI: 10.7150/ntno.77564

DOI: 10.1016/j.pestbp.2020.104586
Conclusions should reaffirm the fundamental contribution of this paper.

Reviewer 4 Report

In this manuscript, Rahman and co-workers reviewed recent progress related to solid lipid nanoparticle (SLN) research. The authors first introduced the background information of SLNs in detail; categorizing pros/cons with bullet points makes it easy for readers to follow. Next, the authors reviewed the delivery mechanism of SLNs, characteristics, and applications of SLNs in cancer. The manuscript is well-organized and easy to follow. SLNs are proven to be an effective platform for drug delivery and cancer treatment. Thus, the reviewer thinks this manuscript can be a nice addition to the field and recommend publication in its current form. 

Author Response

The reviewer accepted the manuscript in the present form.